# Resource-efficient Pure Exploration for Contextual Bandits

## Abstract

It is often of interest to learn a context-sensitive decision policy, such as in contextual multi-armed bandit processes. To quantify the efficiency of a machine learning algorithm for such settings, probably approximately correct (PAC) bounds, which bound the number of samples required, or cumulative regret guarantees, are typically used. However, many real-world settings have a limited amount of resources for experimentation, and decisions/interventions may differ in the amount of resources required (e.g., money or time). Therefore it is of interest to consider how to design an experiment strategy to learn a near-optimal contextual policy while minimizing the total amount of resources required. In contrast to RL or bandit approaches that embed costs into the reward function, here we focus on minimizing the resources needed to learn a near-optimal policy without resource constraints, which is similar to PAC-style approaches which seek to minimize the amount of data needed to learn a near-optimal policy. We propose two resource-aware algorithms for the contextual bandit setting and provide finite sample performance bounds on the resulting best policy that can be obtained from each of the algorithms. We also evaluate both algorithms on synthetic and semi-synthetic datasets and show that they significantly reduce the total resources needed to learn a near-optimal decision policy compared to prior approaches that use resource-unaware exploration strategies.

## 1 Introduction

Contextual multi-armed bandits (CMABs) provide a framework for learning to make the best decision per context. As technology advancements have made it easier to deploy personalized interventions than was possible historically, CMABs are increasingly relevant to a large range of applications including customer recommendations, health interventions, and educational technology (Chapelle & Li, 2011; Liao et al., 2020; Rabbi et al., 2015; Battalio et al., 2021; Bassen et al., 2020; Mu et al., 2021; Henderson et al., 2021). In many settings, it is common practice to run an experiment to evaluate the potential benefit of a new program. Experiments are expensive and different interventions may require different resources (such as time or money). Consequently, it would be helpful to have algorithms that could be used to minimize the experimental budget needed to learn a near-optimal contextual decision policy for future use. We call this resource-efficient pure exploration for CMABs.

Consider designing a program to support people to attend their court date (Fishbane et al., 2020; Chohlas-Wood et al., 2021). Missing a required court appearance can have severe negative consequences even for a minor offense, which many defendants may be unaware of and wastes time and money in the judicial system. Therefore, it is of interest to increase the court appearance rate. Historically, if one wanted to test out interventions that were specific to each individual, one would have had to rely on mailing different interventions to different individuals. This process requires significant manual human effort and provides limited time specificity, as it is expensive or impossible to obtain fine-grained timing. In contrast, as the majority of people now carry a cell phone, one can now automatically send targeted intervention support (such as text messages, transit coupons, or gift certificates) to specific individuals at specific times. In this setting, different interventions carry different costs, and these costs may vary by the individual: for example, a ride to the court house will be more expensive for someone who lives far from the court house, though providing such support may also be more helpful in enabling that individual to come since the barrier to

appear is larger. A natural question is whether we can minimize the amount of cost needed to learn a good context-specific decision policy.

To our knowledge, resource-efficient pure exploration for CMABs has not received prior attention. Historically, the majority of research on multi-armed bandits (MABs) and CMABs has focused on cumulative regret minimization (Lattimore & Szepesvári, 2020). Cumulative regret measures policy suboptimality at each round of interaction during exploration, whereas our pure exploration setting aims to minimize policy suboptimality only at the end of exploration. For simple (i.e., non-contextual) MABs, the problem of identifying the best arm (Audibert et al., 2010; Jamieson & Nowak, 2014; Kaufmann et al., 2016) has received significant attention. Most of this work focuses on sample efficiency and does not consider cost. Another line of work considers knapsack bandits where there is a fixed budget and tries to maximize the reward obtained given that cumulative budget (Slivkins, 2019). The majority of this work focuses on the MAB setting though there is some work for the CMAB case (Wu et al., 2015) which has again focused on cumulative regret given a bound on the total resources used. Recently there have been a few papers on pure exploration for CMABs (Zanette et al., 2021; Li et al., 2022; Krishnamurthy et al., 2023). These papers have not considered when actions may have heterogeneous costs. In the adaptive exploration literature (including Bayesian optimization) there has been some work on cost-aware exploration (Snoek et al., 2012; Lee et al., 2021; Belakaria et al., 2023; Astudillo et al., 2021; Paria et al., 2020), but this work has not considered the CMAB setting and has largely not provided finite sample bounds on the resulting learned optima (though see Paria et al. (2020)).

In the worst case, resource-efficient pure exploration for contextual bandits may be of similar hardness as sample-efficient pure exploration: as pointed out by Chohlas-Wood et al. (2021), if there is no information sharing across contexts and actions, acquiring information about the outcomes of a particular context-action pair will require sampling it directly. However, in many settings of interest additional structure in the context-action space permits information sharing that enables more efficient, resource-aware learning. For example, when increasing court appearance rates, demographic features may impact appearance probabilities regardless of the intervention, and interventions with similar features (consider interventions that vary the number of text messages received) may have related appearance rates. As we will demonstrate empirically, such information sharing can be leveraged to create significantly more resource-efficient learning.

Our paper makes the following contributions:

- We introduce the problem of resource-efficient pure exploration for contextual multi-armed bandits.

- We introduce two algorithms for resource-aware pure exploration in CMABs: (1) a non-adaptive algorithm for settings where the policy cannot be updated during exploration, (2) an adaptive Bayesian algorithm for settings where we can update our policy as exploration proceeds.

- We provide finite sample performance bounds on the resulting best policy that can be obtained from each of these approaches, which immediately implies guarantees that asymptotically these approaches will recover the optimal policy. These bounds are asymptotically equivalent to their resource-unaware counterparts up to a problem-dependent constant that depends on the cost structure.

- More significantly, we empirically demonstrate our resource-aware algorithms can learn a near-optimal decision policy with substantially fewer resources than prior resource-unaware algorithms, on synthetic and semi-synthetic simulations, including a semi-synthetic court appearance simulator (Chohlas-Wood et al., 2021). These findings highlight the potential of resource-aware exploration. Our code and data are in Supplementary Material and will be publicly available.

We conclude the paper with a discussion of open issues.

## 2 Related Work

There is an extensive and growing literature on CMABs and adaptive experimental design.

*Cumulative reward optimization for MABs with a budget constraint.* A number of papers have considered how to adaptively pull arms to maximize cumulative reward given a constraint on the total budget used, where

arms may have heterogeneous treatment effects. These are often called bandits with knapsack constraints (Badanidiyuru et al., 2014; Agrawal & Devanur, 2016; Agrawal et al., 2016b;a; Slivkins, 2019), and most of this work focuses on the non-contextual bandit setting, with some work considering Bayesian approaches (Xia et al., 2015). Sinha et al. (2021) introduce a variant of the MAB problem where the learner is willing to tolerate a small loss from the highest reward to reduce costs. There has been less work on CMABs, and most of this work has focused on theoretical analysis, and the algorithms are not always computationally tractable. For CMABs with budget constraints, Wu et al. (2015) introduce an approximate linear programming method for small discrete state spaces and provide cumulative regret bounds. Another line of work that is loosely related to our setting is conservative bandits (Wu et al., 2016), where the learner maximizes the cumulative reward while ensuring the reward of the chosen arm is above a fixed percentage of a known arm.

*Pure exploration in CMABs.* Recently there have been several papers that focus on quickly learning a good decision policy for CMABs, in the efficient pure exploration setting. Zanette et al. (2021) provide a static (non-adaptive) algorithm with tight minimax bounds on the number of samples needed for learning a contextual policy with expected near-optimal performance in linear CMABs. Li et al. (2022) provide an instance-optimal algorithm for PAC learning of the optimal policy within a policy class for contextual bandits, and very recent work by Krishnamurthy et al. (2023) presents an algorithm for balancing simple regret and cumulative regret minimization. None of these consider settings where actions have heterogeneous costs.

*Active learning.* Our setting is loosely related to the active learning problem in machine learning, where the goal is to maximize the model accuracy while minimizing the total cost of annotating the data used to train the model. Many previous studies assume that the cost of obtaining each sample is the same, some studies consider varying costs (Settles et al., 2008; Kapoor et al., 2007; Haertel et al., 2008). However, active learning is focused on supervised learning models where the next sample is chosen, and the full label is observed. However, in our contextual bandit setting, we do not get to choose the next state, and we only get to choose the action for a given state and observe its reward– in this sense, we are in the partial information setting (we observe no rewards/labels for actions that are not selected).

*Bayesian optimization and Experimental design.* Our setting also overlaps broadly with many other research areas focused on efficient data collection to learn the optima of a function (Bayesian optimization) or to gather as much information as possible about some parameters of interest (Bayesian optimal experimental design). While Bayesian optimization and pure exploration in bandits are closely related (Srinivas et al., 2012; Krause & Ong, 2011), Bayesian optimization techniques often use Gaussian processes to model complex, black-box functions, while bandit algorithms often leverage parametric structure in rewards for statistical efficiency gains. Some Bayesian optimization papers explicitly consider heterogeneous sampling cost in the acquisition function used to direct sampling (Snoek et al., 2012; Lee et al., 2021; Astudillo et al., 2021; Belakaria et al., 2023). A simple approach proposed is to move from the popular acquisition function of expected improvement, to expected improvement per unit of cost (Snoek et al., 2012). Recent work shows this can be suboptimal in the Bayesian optimization setting (Astudillo et al., 2021), and has considered unknown costs and using a multi-step lookahead approach (Astudillo et al., 2021; Lee et al., 2021) with a finite fixed budget. However, this work has focused on learning the optima for generic function optimization, has not considered parametric structure, and are focused on finding the best optima given a fixed input budget. In contrast, we provide finite bounds that can be used to bound the expected simple regret of the learned contextual policy. Perhaps most similar is work by Paria et al. (2020), which uses a cost-aware version of information-directed sampling (Russo & Van Roy, 2018) to guide exploration for generic Bayesian optimization. One of our algorithms is also related to information-directed sampling, but we formulate a different objective and focus on CMABs.

## 3 Setting

We consider the stochastic contextual bandit environment where at each round $n \in [N]$, a context $s_n \in \mathcal{S}$ is sampled i.i.d. from a distribution $\mu$. For each context $s_n$, a (potentially context-dependent) action set $\mathcal{A}_{s_n}$ is made available to the learner. The bandit instance is defined by a reward function $r : \mathcal{S} \times \mathcal{A}_s \mapsto \mathbb{R}$. Upon choosing an action $a_n \in \mathcal{A}_{s_n}$, a stochastic sample $r_n$ with mean $r(s_n, a_n)$ is revealed to the learner. The reward model parameterization will depend on the problem setting and we will shortly consider several

specific settings. We assume there is a known, deterministic, non-negative resource cost $c(s, a) \in \mathbb{R}^+$ for each state-action pair.

We define a decision policy $\pi$ be a mapping from contexts to actions: $\pi : \mathcal{S} \to \mathcal{A}$. Let $V(\pi)$ denote the expected reward (i.e., value) of a policy $\pi$,

$$V(\pi) := \mathbb{E}_{s \sim \mu}[r(s, \pi(s))], \tag{1}$$

where the expectation is taken over the context distribution, the stochasticity in the observed rewards, and any stochasticity in the policy $\pi$. The optimal policy $\pi^\star$ maximizes the expected reward: $\pi^\star = \max_\pi V(\pi)$.

In the pure exploration / simple regret setting, the goal is create an efficient exploration policy $\pi_e$ to gather a dataset $\mathcal{D}$, such that a near-optimal policy $\widehat{\pi}(\mathcal{D})$ can be learned from the resulting dataset $\mathcal{D}$.

In prior work, efficiency has been defined by the number of samples needed to achieve a particular performance bound $\epsilon$ on the resulting policy

$$V(\pi^\star) - V(\widehat{\pi}(\mathcal{D})) \leq \epsilon \tag{2}$$

In our setting, we are interested in designing resource-efficient exploration algorithms, which aims to reduce or minimize the sum of costs incurred during exploration $c(\mathcal{D}) = \sum_{(s_n, a_n) \in \mathcal{D}} c(s_n, a_n)$ relative to the $\epsilon$-accuracy of the resulting learned policy. While provably minimizing this cost may involve complex optimization programs (similar to knapsack problems), we will shortly introduce and show that myopic cost-aware exploration strategies involve the same computational cost as prior related methods, but can offer notable improvements in the cost required to learn the same $\epsilon$-optimal policy.

Finally, in CMABs, the primary focus has been on the realizable setting where we assume access to a statistical parameterized model that can capture the true reward function. We will make a similar assumption, and assume access to a particular function class (such as a linear model) that describes the reward function. We will consider both the frequentist setting where there is a single fixed but unknown parameter and a Bayesian setting in which a prior over the reward model parameters is provided.

In general, exactly optimizing this objective may be very challenging, but fortunately we will shortly see that it is possible to obtain good performance and theoretical guarantees with some simple algorithms. Before proceeding we briefly define our notation.

**Notation.** Unless otherwise stated, we let $\|x\|$ denote the $l_2$-norm of a vector $x \in \mathbb{R}^d$. For a positive semi-definite matrix $\Sigma \in \mathbb{R}^{d \times d}$, let $\|x\|_\Sigma = \sqrt{x^\top \Sigma x}$. For a set $S$ we let $\Delta(S)$ denote the set of (appropriately defined) distributions over $S$. We use $I_d \in \mathbb{R}^{d \times d}$ to denote the $d$-dimensional identity matrix.

## 4 Algorithms

In the pure exploration setting, a key question is whether it is required to specify an exploration policy in advance of data collection (the *static* setting) or whether it is possible to update the exploration policy during data collection in response to observed rewards (the *adaptive* setting). We present two algorithms, one for each setting, that build on prior work by introducing modifications to past algorithms to account for costs. Perhaps surprisingly, we demonstrate that these modifications yield finite data simple regret bounds that are asymptotically equivalent to their resource-unaware counterparts, up to a problem-dependent constant.

### 4.1 Static Pure Exploration

In many practical settings of interest, it is not possible to deploy a policy that is updated during exploration (Zanette et al., 2021). Continual updates can require significant engineering overhead, and may even be infeasible when rewards are delayed or in studies with parallel treatment assignment. Consider educational applications where many students are assigned to different learning conditions in parallel and learning outcomes can only be measured months later. Here, we propose a cost-sensitive algorithm for static exploration in settings where different actions have different costs.

In particular, we restrict ourselves to the well-studied stochastic *linear* contextual bandits setting. We assume there is a known feature map $\phi : \mathcal{S} \times \mathcal{A}_s \mapsto \mathbb{R}^d$ and the reward model follows $r_{\theta^\star}(s, a) = \phi(s, a)^\top \theta^\star$, where $\theta^\star \in \mathbb{R}^d$ is an unknown parameter. Upon choosing an action $a \in \mathcal{A}_s$, the reward $r = r_{\theta^\star}(s, a) + \eta$ is revealed to the learner, where $\eta$ is mean-zero, 1-sub-Gaussian noise. As is standard, we assume that $\|\theta^\star\| \leq 1$ and $\sup_{s,a} \|\phi(s, a)\| \leq 1$ such that $|r_{\theta^\star}(s, a)| \leq 1$. For a given parameter $\theta \in \mathbb{R}^d$, we define $\pi_\theta(s) = \arg\max_{a \in \mathcal{A}_s} \phi(s, a)^\top \theta$ to be greedy policy with respect to $\theta$. The optimal policy $\pi^\star$ is equivalently defined as $\pi_{\theta^\star}$.

Our static pure exploration setting proceeds in two phases. First, we design an exploration policy $\pi_e$ to to construct a dataset $\mathcal{D}' = \{(s'_n, a'_n, r'_n)\}_{n=1,\ldots,N}$. Then, using $\mathcal{D}'$, we extract the regularized least-square predictor $\hat{\theta} = (\Sigma'_N)^{-1} \sum_{i=1}^N \phi(s'_n, a'_n) r'_n$, where $\Sigma_N = \lambda I_d + \sum_{n \in [N]} \phi(s'_n, a'_n) \phi(s'_n, a'_n)^\top$ and $\lambda > 0$. Our objective is to design $\pi_e$ such that the simple regret of the greedy decision policy $\hat{\pi} = \arg\max_{a \in \mathcal{A}_s} \phi(s, a)^\top \hat{\theta}$ is minimized.

Following Zanette et al. (2021), it can be shown that

$$V(\pi^\star) - V(\hat{\pi}) \leq 2\mathbb{E}_{s \sim \mu} \max_a |\phi(s, a)^\top (\theta^\star - \hat{\theta})|. \tag{3}$$

Applying standard concentration inequalties, the following bound holds with probability at least $1 - \delta$,

$$2\mathbb{E}_{s \sim \mu} \max_a |\phi(s, a)^\top (\theta^\star - \hat{\theta})| \tag{4}$$
$$\leq 2\beta_\delta \mathbb{E}_{s \sim \mu} \max_a \|\phi(s, a)\|_{\Sigma^{-1}},$$

where $\beta_\delta = \left( \sqrt{2 \log(2|\mathcal{A}|/\delta)} \wedge \sqrt{8d \log(6/\delta)} \right) + \sqrt{\lambda}$.

Thus, to learn an $\epsilon$-optimal policy for $\epsilon > 0$, it suffices to design the exploration policy $\pi_e$ so as to minimize the maximum uncertainty $\mathbb{E}_{s \sim \mu} \max_a \|\phi(s, a)\|_{\Sigma^{-1}}$.

Zanette et al. (2021) propose the sampler-planner algorithm (S-P), which consists of two subroutines, the *planner* and the *sampler*. The planner leverages an offline set of contexts and runs a reward-free version of the LinUCB algorithm (Abbasi-Yadkori et al., 2011) that, every time a context $s_m \sim \mu$ is observed, selects action $a_m = \arg\max_a \|\phi(s, a)\|_{\Sigma_m^{-1}}$ that maximizes the uncertainty with respect to the current covariance matrix. Upon termination, the planner outputs a sequence of policies $\pi_1, \ldots, \pi_M$. The sampler then uses the average mixture policy $\pi_{mix}$ to gather a dataset: for each new context, $\pi_{mix}$ samples an index $m \in [M]$ uniformly at random and plays $\pi_m$. Through matrix concentration inequalities, the authors demonstrate that the sampler's policy produces a covariance matrix close to what the planner computed with offline data, which in turn yields a bound on maximum uncertainty and thus simple regret.

Building on the S-P algorithm, we propose the S-P_COST algorithm that consists of two subroutines: the *cost-sensitive planner* (see Alg. 1) and *cost-sensitive sampler* (see Alg. 2). The difference between our S-P_COST and S-P is that S-P_COST, every time a context $s_m \sim \mu$ is observed, chooses action $a_m = \arg\max_a \frac{\|\phi(s,a)\|_{\Sigma_m^{-1}}^2}{c(s,a)}$ (see line 9 of Alg. 1). Intuitively, $\frac{\|\phi(s,a)\|_{\Sigma_m^{-1}}^2}{c(s,a)}$ represents the uncertainty per unit cost, and we would like to maximize the uncertainty reduction per unit cost.

## 4.2 Bayesian Resource-Aware Pure Exploration

Bayesian approaches are very popular in adaptive optimization and experimental design, in part because they provide a natural way to quantify information gain with respect to prior uncertainty, which can be leveraged for adaptive exploration. We introduce a simple resource-aware algorithm for pure exploration in Bayesian contextual bandits.

We here consider a more general class of reward models $f$, such that the observed reward when taking action $a$ in context $s$ is $r = f(s, a, \theta) + \eta$, where $\theta$ is the unknown parameter, $\eta$ is mean-zero, 1-sub-Gaussian noise. We assume that $\theta$ is sampled from some known prior distribution.

Let $\mathcal{F}_n = \{s_1, a_1, r_1, s_2, a_2, r_2, \ldots, s_{n-1}, a_{n-1}, r_{n-1})$ be the sequence of states observed, actions taken, and rewards observed up to the current time point. Define $\mathbb{E}_n[X] \coloneqq \mathbb{E}[X|\mathcal{F}_n]$. Recall that the entropy of

---

**Algorithm 1** COST-SENSITIVE PLANNER

---

1: **Input**: Contexts $\mathcal{C} = \{s_1, \ldots, s_M\}$, reg. $\lambda_{reg}$
2: $\Sigma_1 = \lambda_{reg} I$
3: $m = 1$
4: **for** $m = 1, 2, \ldots M$ **do**
5:     **if** $\det(\Sigma_m) > 2 \det(\Sigma_{\underline{m}})$ or $m = 1$ **then**
6:         $\underline{m} \leftarrow m$
7:         $\Sigma_{\underline{m}} \leftarrow \Sigma_m$
8:     **end if**
9:     Define $\pi_m : s \mapsto \mathrm{argmax}_{a \in \mathcal{A}_s} \frac{\|\phi(s,a)\|^2_{\Sigma_{\underline{m}}^{-1}}}{c(s,a)}$
10:     $\Sigma_{m+1} = \Sigma_m + \alpha \phi_m \phi_m^\top; \; \phi_m = \phi(s_m, \pi_m(s_m))$
11: **end for**
12: **return** policy mixture $\pi_{mix}$ of $\{\pi_1, \ldots, \pi_M\}$

---

**Algorithm 2** COST-SENSITIVE SAMPLER

---

1: **Input**: $\pi_{mix} = \{\pi_1, \ldots, \pi_M\}$, reg. $\lambda_{reg}$
2: Set $\mathcal{D}' = \emptyset$
3: **for** $n = 1, 2, \ldots N$ **do**
4:     Receive context $s_n' \sim \mu$
5:     Sample $m \in [M]$ uniformly at random
6:     Select action $a_n' = \pi_m(s_n')$
7:     Receive feedback reward $r_n'$
8:     Store feedback $\mathcal{D}' = \mathcal{D}' \cup \{s_n', a_n', r_n'\}$
9: **end for**
10: **return** dataset $\mathcal{D}'$

---

a probability distribution $P_x$ is defined as $H(P_x) = -\sum_x P(x) \log P(x)$. Given a history $\mathcal{F}_n$, a prior $p(\theta)$, and an observed state $s_n$, we can define a posterior probability distribution over the optimal action $a^\star = \mathrm{argmax}_{a \in \mathcal{A}_{s_n}} f(s_n, a, \theta)$ in state $s_n$:

$$\alpha_n(s_n, a) = P(a^\star = a | s_n, \mathcal{F}_n). \tag{5}$$

The information gain $g_n(a')$ of selecting a particular action $a'$ in state $s_n$ is defined as the expected reduction in entropy over the optimal action for state $s_n$ after taking action $a'$:

$$g_n(a') = \mathbb{E}_n \left[ H(\alpha_n(s_n, \cdot)) - H(\alpha_{n+1}(s_n, \cdot)) \right]. \tag{6}$$

A common approach in Bayesian optimization that can also be easily applied in the pure exploration simple multi-armed bandit setting is to select actions to maximize the information gain. Russo & Van Roy (2018) introduced information-directed sampling for Bayesian cumulative regret minimization in bandits and extend the above by considering the ratio of the expected regret to the information gain.

In our setting, we are instead interested in considering the information gain in relation to resources spent. To do so, we define our exploration policy as one that maximizes the relative information gain per unit cost:

$$\pi_e(s_n) = \arg\max_{a'} \frac{g_n(a')}{c(s_n, a')}. \tag{7}$$

Our objective is very similar to work in Bayesian optimization that uses expected improvement per unit of cost an acquisition function (Snoek et al., 2012), though that work did not consider multi-armed bandits or the contextual setting, nor provided finite sample analysis.

In general, computing the information gain is computationally challenging due to intractable posteriors. Prior work often considers approximations, and we draw from Russo & Van Roy (2018)'s algorithm for

---

**Algorithm 3** IG_COST

1: **Input**: $K, r, q$
2: Set $\mathcal{D}'_0 = \emptyset$
3: **for** $n = 1, 2, \ldots N$ **do**
4:        Receive context $s'_n \sim \mu$
5:        Draw $\theta^1, \ldots, \theta^K$ from the posterior $p(\theta|\mathcal{F}_n)$
6:        $\hat{\Theta}_a \leftarrow \{m | a = \arg\max_{a'} \sum_y q_{\theta^m, s'_n, a'}(y) r(y)\}$
7:        $\hat{p}(a^*) \leftarrow |\hat{\Theta}_{a^*}|/K \qquad \forall a^*$
8:        $\hat{p}_a(y) \leftarrow \sum_m q_{\theta^m, s'_n, a}(y)/K \qquad \forall y$
9:        $\hat{p}_a(a^*, y) \leftarrow \sum_{m \in \hat{\Theta}_{a^*}} q_{\theta^m, s'_n, a}(y)/K \qquad \forall a^*, y$
10:       $\vec{g}_a \leftarrow \sum_{a^*, y} \hat{p}_a(a^*, y) \log \frac{\hat{p}_a(a^*, y)}{\hat{p}(a^*)\hat{p}_a(y)} \qquad \forall a \in \mathcal{A}_{s'_n}$
11:       Select action $a'_n = \arg\max_{a \in \mathcal{A}_{s'_n}} \frac{\vec{g}_a}{c(s'_n, a)}$
12:       Receive feedback reward $r'_n$
13:       Store feedback $\mathcal{D}'_n = \mathcal{D}'_{n-1} \cup \{s'_n, a'_n, r'_n\}$
14: **end for**
15: **return** dataset $\mathcal{D}'_N$

---

a sample-based approximation to the information gain and present a cost-sensitive information gathering algorithm for contextual bandits in Alg. 3. This uses a sample-based approximation to Equation 7 and we restrict our attention to settings with discrete reward outcomes. We let $y(s, a) \in \mathcal{Y}$ denote the outcome of choosing action a in context s, where $\mathcal{Y}$ is a discrete set.

Alg. 3 takes as input $K, r$, and $q$. $K$ is the number of samples drawn independently from the posterior $p(\theta|\mathcal{F}_n)$ and $r : \mathcal{Y} \mapsto \mathbb{R}$ is a reward function mapping outcomes to scalar rewards. We let $q_{\theta, s, a}(y) = P(y(s, a) = y|\theta)$ be the probability, conditioned on $\theta$, of observing $y$ when action $a$ is selected in context $s$. Line 6 computes the optimal action for each value of $\theta$. Line 7 computes the probability that each action is optimal. Line 8 computes the marginal distribution over the particular rewards, and line 9 computes the joint probability distribution of the optimal action and a particular reward outcome. These quantities are used to compute the information gain (see a derivation in Appendix A.1), which is then scaled by the inverse of the cost.

## 5 Experiments

### 5.1 Synthetic dataset

We first conduct a synthetic experiment to demonstrate the performance of S-P_COST (Alg. 1 & 2) using a simulator inspired by Zanette et al. (2021). We construct a simple linear contextual bandit problem with $d = 10$ and $\mathcal{A} = \{1, \ldots, 20\}$. Each context $s \in \mathcal{S}$ is associated with features vectors $\{\phi(s, a)\}_{a \in \mathcal{A}}$. Each context belongs to one of two discrete categories with equal probability. In category 1, the action $a \in \{1, \ldots, 10\}$ has features distributed as $\phi(s, a) \sim \mathcal{N}(0, \Sigma_a)$ where $\Sigma_a = \text{diag}(x_1, \ldots, x_d)$ with $x_a = 1$ and $x_i = 0$ for all $i \neq a$. The action $a \in \{11, \ldots, 20\}$ for category 1 has features distributed as $\phi(s, a) \sim \mathcal{N}(0, \Sigma_a)$ where $\Sigma_a = \text{diag}(x_1, \ldots, x_d)$ with $x_a = 1$ and $x_i = 10^{-9}$ for all $i \neq a$. In category 2, the action $a \in \{1, \ldots, 19\}$ shares the same feature distribution as $a \in \{1, \ldots, 19\}$ in category 1, and the last action $a = 20$ has features distributed as $\phi(s, 20) \sim \mathcal{N}(0, \text{diag}(2, 10^{-9}, \ldots, 10^{-9}))$. $c(s, a) = \|\phi(s, a)\|_0, \forall s, a$. The first $d - 1$ coordinates of $\theta^\star$ are 1, and the last coordinate is 0. The linear reward model is defined as $r_{\theta^\star}(s, a) = \phi(s, a)^\top \theta^\star + \eta$ for some $\theta^\star \in \mathbb{R}^d$ parameter and some mean-zero 1-subgaussian random variable $\eta$. The synthetic dataset is designed in such a way that about half of the actions are slightly more informative but cost much more than the other half.

We compare our S-P_COST against two baselines: (1) a random exploration algorithm (RANDOM) that chooses actions uniformly and (2) the cost-unaware sampler-planner algorithm (S-P) proposed in Zanette et al. (2021). For S-P and S-P_COST, the planner is first run on an independent set of contexts of size $M = 500$. All algorithms are then used to collect a dataset of size $N = 500$. After each sample is collected

during exploration, we calculate the value of the resulting greedy decision policy for each algorithm on a held-out test set of 500 data points. The policy value is computed as the average reward on the test set. Additionally, we calculate the value of the optimal policy, which has access to the true reward model, as a baseline. All algorithms used $\lambda = 1$, and the planner used $\alpha = 1$, as these work well empirically (Zanette et al., 2021). We ran the experiment for 50 trials using random seeds 1-50.

We find that S-P_cost learns an $\epsilon$-optimal policy at a lower exploration cost than S-P. Figure 1a shows cumulative exploration costs plotted over the performance gap $\epsilon$ (i.e., the difference between the value of the optimal policy and the learned policy). This is because S-P_cost tends to sample actions $1 - 10$ that have a cost of 1 in both contexts, which is sufficient to get information about all the coordinates, while S-P tends to sample actions $10 - 20$ and suffers a cost of 10 for each sample.

## 5.2 Court appearance dataset

We evaluate IG_cost using a semi-synthetic court appearance simulator from Chohlas-Wood et al. (2021), which is grounded in real case data from the Santa Clara County Public Defender Office. In this setting, a policymaker seeks to help individuals attend their mandatory court dates by providing government-sponsored transportation assistance. Individuals can receive one of three mutually exclusive interventions $a$: rideshare assistance, a transit voucher, or no transportation assistance. The round-trip rides cost \$5 for every mile between an individual's home address and the main courthouse and back. The transit voucher costs \$7.5. Since our algorithm requires $c(s, a) > 0, \forall s, a$, we assume that the no transportation assistance intervention has a cost of \$0.1, which is negligible compared to the other interventions. The simulator considers the binary outcome $y \in \{0, 1\}$ that indicates whether a client appeared at their court date. The simulator uses a logistic reward model where $\mathbb{P}(r_{\theta^\star}(s, a) = 1) = \text{logit}^{-1}(\phi(s, a)^\top \theta^\star)$ for some unknown $\theta^\star \in \mathbb{R}^d$. The reward is independent across draws.

The resulting dataset consists of $12,636$ example cases. Each data point is a 7-dimensional feature vector associated with the true appearance probability of the individual, the observed binary outcome, and the cost of the intervention if provided each of the three interventions. The simulator is designed in such a way that the type of assistance that is best for each individual varies across the population. The goal is to learn which intervention maximizes the appearance probability for each individual.

We compare our IG_cost against four baselines: (1) a random exploration algorithm (RANDOM) that always chooses actions uniformly, (2) Thompson sampling (THOMPSON) (Russo et al., 2018), (3) Contextual Gaussian Process Upper Confidence Bound (CGP-UCB) (Krause & Ong, 2011), and (4) the cost-unaware algorithm that only considers the information gain (IG). THOMPSON maintains a posterior over the parameters of the reward model. Every time a context $s \sim \mu$ is observed, THOMPSON samples $\theta$ from this posterior and selects the action $a = \arg\max_a r_\theta(s, a)$. We compare against THOMPSON because prior work (Chohlas-Wood et al., 2021) has shown that THOMPSON is good at simple regret minimization even though it optimizes for cumulative regret. CGP-UCB is a Bayesian optimization approach to pure exploration that relies on an underlying Gaussian process (GP) that takes as input a policy's parameters and is trained to predict policy value. CGP-UCB proceeds by selecting a candidate policy that maximizes the upper confidence bound over policy value as predicted by the GP model. We also introduce IG, which selects action $a = \arg\max \vec{g}_a$ that maximizes the information gain instead of the relative information gain per cost (see line 11 of Alg. 3). Following Chohlas-Wood et al. (2021), we use non-informative priors, and we use the `sim` function in `arm` (Gelman, 2011) to do posterior sampling (see experiment details Appendix A.2).

We run the experiment for 50 trials using random seeds 1-50. In each trial, we randomly sample 1500 data points as the training data and test data, respectively. Following Chohlas-Wood et al. (2021), we start each trial with a randomly selected warm-up group of 4 people. We run all algorithms on the training data such that they observe the same contexts but may take different actions. After each training observation, we calculate the value of the resulting greedy decision policies, which is the appearance rate under the policy on test data. As a baseline, we calculate the value of the optimal policy given access to the true reward model.

We find that IG_cost can learn an $\epsilon$-optimal decision policy using substantially fewer exploration resources than the other cost-unaware algorithms. Figure 1b shows cumulative exploration costs plotted over the

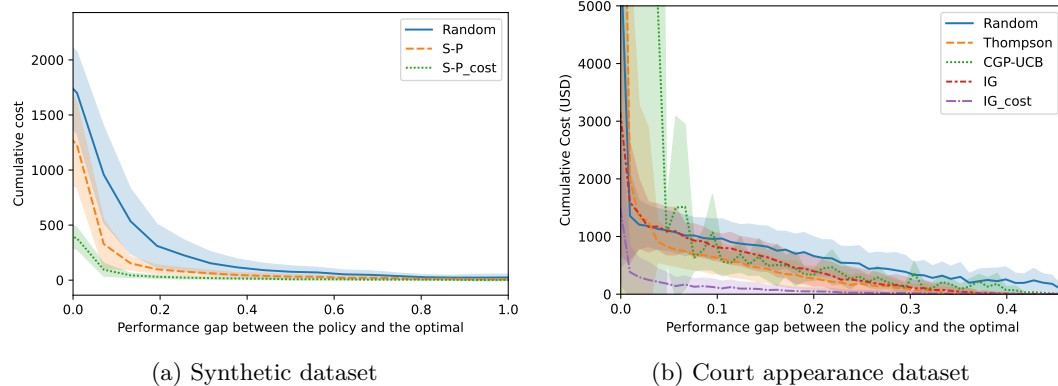

(a) Synthetic dataset  (b) Court appearance dataset

Figure 1: Our cost-aware algorithms (S-P_COST and IG_COST) can learn a near-optimal decision policy using substantially fewer resources than traditional cost-unaware algorithms in both the synthetic and court appearance settings. The x-axis shows the performance gap $\epsilon$, which is the difference between the value of the optimal policy and the learned decision policy. The y-axis shows the total cost of the experiment. Each line represents the mean of 50 trials, with error bars indicating the standard deviation. We averaged costs within 50 equally spaced $\epsilon$ intervals. For (b), we smoothed the data using a 30-iteration running average.

performance gap $\epsilon$ (i.e., the difference between the appearance rate under the optimal policy and the learned policy).

## 6  Theoretical Guarantees

In this section, we study the theoretical properties of our cost-sensitive algorithms and show that they can provably recover $\epsilon$-optimal policies in certain settings, provided the ratio between the maximum costs and minimum costs is bounded. Our objective is to bound the difference in values between the optimal policy $\pi^\star$ and the learned policy $\hat{\pi}$ in terms of the number of interactions. In particular, we find that the cost-sensitive algorithms are competitive with cost-unaware algorithms in terms of sample complexity. We define the worst-case cost ratio as $\gamma := \frac{\max_a c(a)}{\min_a c(a)}$. Our first result concerns the S-P_COST algorithm, which, in the stateless case, reduces to selecting $a_n = \mathrm{argmax}_a \frac{\|\phi(a)\|_{\Sigma_n}}{c(a)}$.

**Theorem 6.1** (Cost-sensitive sampler-planner)**.** *For a linear contextual bandit with known feature map* $\phi : (s,a) \mapsto \phi(a)$, *S-P_COST selects* $a_1, \ldots, a_N$ *where* $a_n = \max_a \frac{\|\phi(s_n,a)\|^2_{\Sigma_m^{-1}}}{c(s_n,a)}$. *If we let* $\alpha = 1$, $M = \Theta(N)$, *and* $\lambda = \widetilde{\Omega}(d)$, *then the following holds with probability at least* $1 - \delta$.

$$V(\pi^\star) - V(\hat{\pi}) \leq \widetilde{O}\left(\beta_\delta \sqrt{\frac{\gamma d \log{(\lambda + N)}}{N}}\right). \tag{8}$$

Next, we consider the cost-sensitive information gathering algorithm, IG_COST. Here we provide a generic Bayesian-style analysis where we consider a prior reward functions $r_*$ (and thus $\pi^\star(s) = \mathrm{argmax}_a r_*(s,a)$). Recall the algorithm selects $a_n = \mathrm{argmax}_a \frac{g_n(s_n,a)}{c(s_n,a)}$ where $g_n(s,a)$ is the *information gain*, defined as the conditional mutual information $\mathbb{I}(a_n^*; r_{n,a}|\mathcal{F}_{n-1},s)$ between the reward $r_{n,a}$ that would be received for selecting action $a$ in state $s$ and $a_n^*$ at step $n$, conditioned on the history $\mathcal{F}_{n-1} = (s_1, a_1, r_1, \ldots, s_{n-1}, a_{n-1}, r_{n-1})$ of actions and observations so far. Note that this is the mutual information conditioned on the fixed values $(\mathcal{F}_{n-1}, s)$, not averaged over them.

A key quantity in our analysis is the information ratio, originally introduced by Russo & Van Roy (2018):

$$\Psi_n := \min_\pi \mathbb{E}_s \frac{\mathbb{E}_{a \sim \pi(\cdot|s)}\left[r_*(s,a^*) - r_*(s,a)|\mathcal{F}_{n-1}\right]^2}{\mathbb{E}_{a \sim \pi(\cdot|s)} g_n(s,a)} \tag{9}$$

We assume that $\Psi_n \leq \Psi$ for some $\Psi \in \mathbb{R}_{\geq 0}$. $\Psi$ is known to bounded for many standard bandit problems (Russo & Van Roy, 2018), including the linear contextual bandit, for which we also provide a specialized bound.

**Theorem 6.2** (Cost-sensitive information gathering). *For a contextual bandit problem, IG_COST produces a policy $\hat{\pi}$ over $N$ rounds such that the Bayesian regret is bounded as*

$$\mathbb{E}\left[V(\pi^\star) - V(\hat{\pi})\right] \leq O\left(\sqrt{\frac{\gamma \Psi \mathbb{I}(\pi^\star; \mathcal{F}_N)}{N}}\right), \tag{10}$$

*where the expectation is over the prior of contextual bandit problems, and $\mathbb{I}(\pi^\star; \mathcal{F}_T)$ denotes the mutual information between the optimal policy and the history under the data collection algorithm. Furthermore, if the prior is over $d$-dimensional linear contextual bandits, the Bayesian regret can be bounded as*

$$\mathbb{E}\left[V(\pi^\star) - V(\hat{\pi})\right] = \widetilde{O}\left(\sqrt{\gamma d^2/N}\right). \tag{11}$$

The first applies generally to contextual bandit problems, even nonlinear ones. However, to make concrete the guarantees, especially in well studied settings, we specialize the second bound to the case of linear contextual bandits, which follows by applying prior work (Hao et al., 2022) to bound $\Psi = O(d)$ and $\mathbb{I}(\pi^\star; \mathcal{F}_N) = \widetilde{O}(d)$. This show IG_COST is competitive with standard information-directed sampling that ignores the cost, up to scaling by $\gamma$.

The above two theorems prove that, assuming the ratio between the minimum and maximum cost per context is bounded $\gamma$, we can guarantee that our proposed approaches will converge to an $\epsilon$-optimal policy, and have a sample complexity that is upper bounded with the same dominant terms as prior state-of-the-art methods, with an additional factor of $\sqrt{\gamma}$ (see full proofs in Appendix B).

Naturally, it would be of significant additional interest if we could guarantee that the resource cost complexity, to learn a near-optimal decision policy, from our new methods is guaranteed to be smaller than prior methods. Unfortunately this is, in general, non-trivial and comparisons are subtle due to the granular nature of uncertainty refinement. To illustrate this, consider using our S-P_COST versus prior approach S-P on a non-contextual bandit with a one-dimensional (scalar) action space, with two potential actions: $a_1 = 1.0$ with cost $c(a_1) = 1.0$ and $a_2 = 0.9$ with cost $0.6$. Assume both algorithms use a regularization of $\lambda = 1$. On the first round S-P will select action $a_1$, and S-P_COST will select $a_2$. After this the new covariance matrix $\Sigma_{\text{S-P}} = \lambda + a_1^2 = 2$ and $\Sigma_{\text{S-P\_COST}} = \lambda + a_2^2 = 1.81$. Recall that the covariance matrix (here of the sampled actions) directly controls an upper bound on the resulting policy suboptimality through $\mathbb{E}_s \max_a \|\phi(s, a)\|_{\Sigma^{-1}}$. It is clear that achieving $\Sigma = 1.81$ costs less than achieving $\Sigma = 2$. However, it is not possible to obtain $\Sigma = 2$ using the S-P_COST algorithm, since the next action it will select yields an updated $\Sigma$ of 2 or 2.62. In general, selecting actions will cause the covariance matrix to update in discrete jumps, and so it will be hard to compare the relative cost to achieve a particular $\epsilon$-accuracy, since that might only be feasible by overpaying and achieving lower $\epsilon$ than what was required.

However, it is important to note that there are some cases where S-P_COST will be guaranteed to have smaller cost than S-P. For example, assume that for each state, there always exists 2 actions, one with negligibly smaller norm with cost cmin and the other with cost cmax. Here the updated covariance matrix after each step will be essentially identical, but S-P_COST will cost $O(\frac{c_{min} d^2}{\epsilon^2})$ compared to S-P's cost of $O(\frac{c_{max} d^2}{\epsilon^2})$.

## 7 Discussion and Conclusion

Our approach for Bayesian pure exploration for contextual bandits maximized the information gain per cost for the current context but did not consider the potential benefit for all contexts. Recent work by Hao et al. (2022) has demonstrated that such conditional strategies may be outperformed by context-aware strategies in cumulative regret minimization (without costs), which presents an interesting direction for further investigation. A very interesting open question is whether it is important to consider longer horizons when designing sampling strategies to learn near-optimal policies given heterogeneous resource costs.

Our work has highlighted the substantial potential impact of considering costs during pure exploration. We presented two algorithms that provided substantial gains in illustrative synthetic and semi-synthetic problems and a theoretical analysis that shows that maximizing information gain per cost can still provide simple regret bounds of similar rates to the cost-free case in some settings.

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

# A Algorithm and experiment details

## A.1 Definition of Information Gain in Alg. 3

Let $A^*$ be a random variable that represents the optimal action, and $Y_{n,a}$ be the outcome when action $a$ is selected at time $n$. As in Russo & Van Roy (2018), the information gain from an action $a$ is defined to be:

$$g_n(a) := I_n(A^*; Y_{n,a}) \tag{12}$$

$$= D_{KL}\left(\mathbb{P}\left((A^*, Y_{n,a}) \in \cdot | \mathcal{F}_n\right) \| \mathbb{P}\left(A^* \in \cdot | \mathcal{F}_n\right) \mathbb{P}\left(Y_{n,a} \in \cdot | \mathcal{F}_n\right)\right) \tag{13}$$

$$= \sum_{a^*, y} \hat{p}_a(a^*, y) \log \frac{\hat{p}_a(a^*, y)}{\hat{p}(a^*) \hat{p}_a(y)}. \tag{14}$$

## A.2 Experiment Details for Court Appearance Dataset

In the court appearance dataset, we have 7 raw features for each individual: (1) whether the client identifies as Vietnamese; (2) whether the case is a felony; (3) whether the client identifies as male; (4) the client's age; (5) the natural log of the distance, in miles, between the client's home address and the courthouse, minus the natural log of the maximum allowed distance of 20 miles (so that all distance attributes are negative, with values of higher magnitude being closer to the courthouse); (6) the number of known failures to appear in the past two years; and (7) the inverse number of required court appearances in the past two years. For additional details about the data generation process, see Appendix E in Chohlas-Wood et al. (2021).

Following Chohlas-Wood et al. (2021), we start each trial with a randomly selected warm-up group of 4 people. The first two people are assigned a transit voucher and a rideshare, respectively. The other two people are assigned no transportation assistance. The costs during this warm-up period are not considered in the evaluation. We use non-informative priors for THOMPSON, IG, and IG_COST, and we use the `sim` function in `arm` (Gelman, 2011) to do posterior sampling.

For IG and IG_COST, we use $K = 10$ because a moderate number of samples might be enough to generate close approximations to the information gain (Russo & Van Roy, 2018).

For CGP-UCB, we use the Upper Confidence Bound (UCB) acquisition function with $beta = 0.1$. In each optimization step, we draw 50 policies from the parameter space, and the candidate policy with the highest UCB is evaluated on 5 randomly selected contexts. We estimate the value of the candidate policy using the average reward achieved by the policy on these 5 data points. We update the GP posterior using this 5-sample estimate of $V(\pi)$ at each iteration. Since CGP-UCB requires the search space to be bounded, we constrain the parameter space to $[-5, 5]^d$ based on the true reward parameter. We initialize the Gaussian process model using the greedy decision policy extracted from the warm-up data.

# B Proofs of Theoretical Results

## B.1 Cost-Sensitive Sampler-Planner

Define the following notation as used in the sampler-planner paper:

$$\Sigma_m = \lambda I + \alpha \sum_{s \in [m-1]} \phi_s \phi_s^\top \tag{15}$$

$$\Sigma'_n = \lambda I + \sum_{s \in [m-1]} \phi'_s {\phi'_s}^\top \tag{16}$$

are the covariance matrices collected during the offline and online phases, respectively. Here $\alpha \in (0, 1]$. We also use $\Sigma_{\bar{m}}$ to denote the covariance matrix that is used at round $m$ during batching (since the covariance matrices are only updated once the determinant reaches a certain threshold). The following is a standard error bound on the value difference of the policy learned through the algorithm.

**Lemma B.1.** *The simple regret is bounded in terms of the regression error as*

$$V(\pi^\star) - V(\hat{\pi}) \leq 2\|\theta_\star - \hat{\theta}_N\|_{\Sigma_N'} \cdot \mathbb{E}_s \left[ \max_a \|\phi(s,a)\|_{\Sigma_N'^{-1}} \right]. \tag{17}$$

Standard concentration inequalities will guarantee a bound on $\|\theta_\star - \hat{\theta}_N\|_{\Sigma_N'}$ of order $\widetilde{\mathcal{O}}(\sqrt{d})$. Thus we are concerned with ensuring that $\|\phi(x,a)\|_{\Sigma_N'^{-1}}$ shrinks. We will leverage the following relationship between the planner and sampler, which shows that $\mathbb{E}_s \max_a \|\phi(s,a)\|_{\Sigma_N'^{-1}}$ can be bounded by $\mathbb{E}_s \max_a \|\phi(s,a)\|_{\Sigma_M^{-1}}$.

**Lemma B.2** (Lemma 3 and 4 of Zanette et al. (2021))**.** *Let $K$ be a random variable that denotes the total number of policy switches. Fix $\delta > 0$ and let $\lambda = \Omega(\log(d/\delta))$ and $M = \Omega\left(\frac{KN}{\lambda} \log\left(dNK/\lambda\delta\right)\right)$ and $1/\alpha = \Omega\left(\frac{K}{\lambda} \log\left(dNK/\lambda\delta\right)\right)$. There is an absolute constant $C > 0$ such that, with probability at least $1 - \frac{3}{4}\delta$,*

$$\mathbb{E}_s \max_a \|\phi(s,a)\|_{\Sigma_N'^{-1}} \leq \frac{C}{M} \left( \log(1/\delta) + \sum_{m \in [M]} \max_a \|\phi(s_m,a)\|_{\Sigma_m^{-1}} \right). \tag{18}$$

Note that $K = O(d \log\left(1 + M/d\lambda\right))$ (Lemma 15 of Zanette et al. (2021)), so we can choose $\lambda, M, \alpha$ with enough margin ahead of time.

We note that this result holds as long as the conditions above are satisfied and as long as the algorithms are employed such that the actions taken at sampling time reflect the actions taking during planning (as in sampling from the set of policies that is generated).

*Proof of Theorem 6.1.* We can now show a bound on the average of uncertainties under the planner.

$$\mathbb{E}_s \max_a \|\phi(s,a)\|_{\Sigma_M^{-1}} \leq \frac{C}{M} \sum_m \max_a \|\phi(s_m,a)\|_{\Sigma_{\bar{m}}^{-1}} + \frac{C\log(1/\delta)}{M} \tag{19}$$

$$= \frac{1}{M} \sum_m \max_a \|\phi(s_m,a)\|_{\Sigma_{\bar{m}}^{-1}} \cdot \sqrt{\frac{c(s_m,a)}{c(s_m,a)}} + \frac{C\log(1/\delta)}{M} \tag{20}$$

$$\leq \frac{1}{M} \sum_m \sqrt{c_{\max}(s_m)} \max_a \frac{\|\phi(s_m,a)\|_{\Sigma_{\bar{m}}^{-1}}}{\sqrt{c(s_m,a)}} + \frac{C\log(1/\delta)}{M} \tag{21}$$

$$= \frac{1}{M} \sum_m \sqrt{c_{\max}(s_m)} \frac{\|\phi(s_m,a_m)\|_{\Sigma_{\bar{m}}^{-1}}}{\sqrt{c(s_m,a_m)}} + \frac{C\log(1/\delta)}{M} \tag{22}$$

$$\leq \frac{\sqrt{2}}{M} \sum_m \sqrt{c_{\max}(s_m)} \frac{\|\phi(s_m,a_m)\|_{\Sigma_m^{-1}}}{\sqrt{c(s_m,a_m)}} + \frac{C\log(1/\delta)}{M} \tag{23}$$

$$\leq \frac{\sqrt{2\gamma}}{M} \sum_m \|\phi(s_m,a_m)\|_{\Sigma_m^{-1}} + \frac{C\log(1/\delta)}{M} \tag{24}$$

$$= \widetilde{O} \left( \sqrt{\frac{\gamma dM \log(\lambda + M)}{\alpha}} + C\log(1/\delta) \right). \tag{25}$$

The first inequality uses the above lemma. The third line upper bounds with the worst case action. The fourth line uses the fact that $a_m$ is selected by taking $a_m = \text{argmax} \frac{\|\phi(s_m,a)\|_{\Sigma_{\bar{m}}^{-1}}^2}{c(s_m,a)}$. The fifth line converts the batched covariance matrix to the individual per-step covariance matrices (Abbasi-Yadkori et al., 2011). Uses the upper bound on the ratio between the largest and smallest values of $c$. Finally, we apply the elliptical potential lemma (Lattimore & Szepesvári, 2020). □

## B.2 Cost-Sensitive Information Gathering

*Proof of Theorem 6.2.* As is conventional for analyses of information-directed sampling, we define

$$\Delta_n(s,a) = \mathbb{E}_n \left[ r(s, \pi^\star(s)) - r(x, \hat{\pi}_n(s)) \right] \tag{26}$$

where $\hat{\pi}_n(s) = \mathrm{argmax}_a\, \mathbb{E}_n\left[r(s,a)\right]$ and $\mathbb{E}_n$ is expectation conditioned on information observed up until time $n$.

$$\mathbb{E}\left[V(\pi^\star) - V(\hat{\pi})\right] \le \mathbb{E}\left[\min_a \Delta_N(s,a)\right] \tag{27}$$

$$\le \frac{1}{N}\sum_n \mathbb{E}\left[\min_a \Delta_n(s_n,a)\right] \tag{28}$$

Note that from the choice of $a_n$, we also have

$$\Delta_n(s_n,a) = \frac{\Delta_n(s_n,a)}{\sqrt{g_n(s_n,a_n)/c(s_n,a_n)}} \cdot \sqrt{\frac{g_n(s_n,a_n)}{c(s_n,a_n)}} \tag{29}$$

$$= \frac{\Delta_n(s_n,a)}{\max_{a'}\sqrt{g_n(s_n,a')/c(s_n,a')}} \cdot \sqrt{\frac{g_n(s_n,a_n)}{c(s_n,)}} \tag{30}$$

$$\le \sqrt{\gamma} \cdot \frac{\Delta_n(s_n,a)}{\max_{a'}\sqrt{g_n(s_n,a')}} \cdot \sqrt{g_n(s_n,a_n)} \tag{31}$$

where in the second line, we have used the definition of $a_n$ and the third uses bounded ratio of the max and min costs. Taking the min over $a$, we can bound this with the information ratio:

$$\frac{\min_a \Delta_n(s_n,a)}{\max_{a'}\sqrt{g_n(s_n,a')}} \le \frac{\mathbb{E}_{a\sim\pi(\cdot|s_n)}\left[\Delta_n(s_n,a)\right]}{\sqrt{\mathbb{E}_{a\sim\pi(\cdot|s_n)}g_n(s_n,a)}} \quad \forall \pi \tag{32}$$

which implies that $\mathbb{E}_{s_n}\left[\frac{\min_a \Delta_n(s_n,a)}{\max_{a'}\sqrt{g_n(s_n,a')}}\right] \le \sqrt{\Psi}$. Therefore, we can bound the average of the costs now using Holder's inequality and the information ratio. For simplicity, let $\mathbb{E}_N$ also denote the uniform expectation over the time indices $n \in \{1,\ldots,N\}$. Then,

$$\mathbb{E}\left[V(\pi^\star) - V(\hat{\pi})\right] \le \mathbb{E}_N\left[\min_a \Delta_n(s_n,a)\right] \tag{33}$$

$$\le \sqrt{\gamma} \cdot \mathbb{E}_N\left[\min_\pi \frac{\Delta_n(s_n,\pi)}{\sqrt{g_n(s_n,\pi)}} \cdot \sqrt{g_n(s_n,a_n)}\right] \tag{34}$$

$$\le \sqrt{\gamma \mathbb{E}_N\left[\min_\pi \frac{\Delta_n^2(s_n,\pi)}{g_n(s_n,\pi)}\right] \cdot \mathbb{E}\left[g_n(s_n,a_n)\right]} \tag{35}$$

$$\le \sqrt{\gamma \Psi \frac{1}{N}\mathbb{E}\sum_n g_n(s_n,a_n)} \tag{36}$$

By the data processing inequality and the tensorization of the mutual information, we have that

$$\mathbb{E}\sum_n g_n(s_n,a_n) \le \mathbb{I}(\pi^\star;\mathcal{F}_N). \tag{37}$$

.

We now specialize our result to the linear contextual bandit setting. As per the bound in Equation 36, the key steps will be to bound $\Psi$ and $I(\pi^\star;\mathcal{F}_N)$ for the linear contextual bandit setting. Our results draw strongly from the results presented in Hao et al. (2022).

We first bound $I(\pi^\star;\mathcal{F}_N)$:

$$I(\pi^\star;\mathcal{F}_N) \le I(\theta_*;\mathcal{F}_N), \tag{38}$$

using a second application of the data processing inequality.

We then apply Lemma 5.8 from Hao et al. (2021) that proves $I(\theta_*;\mathcal{F}_N) \le 2d\log(Cdn)$ for some constant $C > 0$.

Next, to bound $\Psi$, we follow the proof structure from Lemma 6.4 of Hao et al. (2022). Specifically, they prove that

$$\mathbb{E}_{s_n} \min_{\pi} \frac{\Delta_n^2(s_n, \pi)}{g_n(s_n, \pi)} \tag{39}$$

is bounded by the information ratio of the Thompson Sampling algorithm, which is bounded by $O(d)$.

Combining these results, this guarantees that the simple regret is bounded by at most $\widetilde{O}\left(\sqrt{\gamma d/N}\right)$. $\qquad \square$

