# OpenReview forum: "Resource-efficient Pure Exploration for Contextual Bandits"
_TMLR — Rejected by TMLR_

### Review · Reviewer_aqEo · 2024-06-06

**Summary Of Contributions:**

This paper formulates the cost-aware contextual multi-armed bandit problem. This paper provides 2 algorithms aiming at minimizing the value function suboptimality error for static and adaptive exploration policies. This paper also provides empirical results, showcasing improvements from prior works.

**Audience:**

Yes

**Broader Impact Concerns:**

As this paper uses and discusses about the "court appearance" problem, it would require a section on Broader Impacts. I understand this work is of initial findings and theoretical in nature, but it is important to mention and acknowledge the same.

**Claims And Evidence:**

Yes

**Requested Changes:**

na

**Strengths And Weaknesses:**

Strengths:

This is a well polished paper.

This paper's contribution sits well in the constrained optimization sub-field of ML.

Weaknesses:

This problem is also related to the more general setting of "Constrained Markov Decision Processes" (lots of references starting E. Altman's book on the same name). Please include more discussions on this with references like [1] etc (this is a random reference I found, not a representative one, so you can choose to refer a different one in this context.)

At the end of Section 4.1, please provide the intuition of the choice of $1/c(s,a)$ for the proposed algorithms.

[1] Sun, Hao, et al. "Safe exploration by solving early terminated mdp." arXiv preprint arXiv:2107.04200 (2021).

Minor typos:

- "RL" in the abstract is abbreviated all along. Please write "reinforcement learning (RL)"
- Missing "The sixth line uses ..." while pointing out the equation in the proof of Thm.6.1

---

> ### Author Response · Authors · 2024-06-21
> **Response**
>
> - We appreciate the reviewer’s encouraging remarks about our paper’s contribution.
> - Constrained Markov Decision Processes:
>   - Thank you for highlighting this. The original work on constrained MDPs (e.g. Altman’s book) focuses on planning with constrained MDPs; in contrast, we are doing online learning of a policy through active data collection for contextual MAB. There is additional work in learning in constrained MDPs such as “A Best-of-Both-Worlds Algorithm for Constrained MDPs with Long-Term Constraints Jacopo Germano, Francesco Emanuele Stradi, Gianmarco Genalti, Matteo Castiglioni, Alberto Marchesi, Nicola Gatti”.  To our knowledge, such work often assumes there is a fixed constraint that must be satisfied, and constrains what policy can be optimal: for example,  “Safe Exploration by Solving Early Terminated MDP“ more efficiently learns a policy that matches the corresponding constrained MDP.  In contrast, in our work we aim to find a near-optimal policy (with no constraints) but reduce the cost required to learn that policy. We will expand our discussion of these points in the paper.
> - "At the end of Section 4.1 please provide the intuition of the choice of of $1/c(s,a)$ for the proposed algorithms":
>   - We are happy to clarify the motivation behind our algorithms. We chose to divide the information gain by cost because it effectively quantifies the efficiency of information acquisition in terms of resource expenditure.
> - Thank you for catching these typos: we will fix them.
> - Broader Impact discussion: We appreciate the useful feedback and will include additional discussion on broader impacts.

---

> > ### Comment · Reviewer_aqEo · 2024-06-24
> >
> > I understand this is different from CMDPs even under the planning setting. But I am not sure if this resource-efficient setting can be thought of as a special case of CMDPs. If so, are these learning algorithms somehow better than constrained policy optimization or other constrained RL algorithms?
> > If not, please do include the planning literature under your setting of reducing costs.
> > I think all concerns are minor at this point.
> > Thanks, good luck with the decisions.

---

> > > ### Author Response · Authors · 2024-06-29
> > > **Response**
> > >
> > > - Thank you for your feedback. Yes, we believe our algorithms offer distinct advantages over other constrained RL algorithms and we've outlined some of those in our responses to other reviewers. In particular, the conservative / safe bandits literature has primarily focused on minimizing cumulative regret, in MABs and CMABs, subject to some constraints on the policy class. For example, Amani et al 2019 instead consider cumulative regret with respect to the best safe actions, whereas we focus on reducing cost while learning a policy that minimizes simple regret. In Pacchiano et al. 2021 they do consider cost directly, but focus on minimizing cumulative regret while ensuring the deployed policy on each round satisfies a cost constraint, compared to our interest in minimizing cumulative cost to learn a good policy that minimizes simple regret. Similar to us, Carlsson et al 2024 focus on learning a near-optimal policy, but they consider multi-armed bandits with no context, and assume there are constraints on the arms. Work by Jamieson and others has considered how in some settings contextual MAB may be reduced to linear MABs, and an interesting direction for future work would be to consider if such results might allow a closer connection with our work (which considers contexts) and Carlsson et al. Note however that they focus on linear constraints on the policies themselves (through arm constraints) whereas we are interested in reducing cumulative cost.
> > > - We will include this discussion in our main paper. We will also include a discussion of the planning setting.

---

### Review · Reviewer_e2SZ · 2024-06-08

**Summary Of Contributions:**

This paper examines a specific type of pure exploration issue in contextual bandits, focusing on reducing sampling costs while meeting certain performance requirements. It introduces two heuristic methods, one static frequentest one (based on sampler-planner procedure) and another for bayesian adaptive scenario (based on information-directed sampling). Numerical evidence demonstrates improvements over baselines, and some theoretical guarantees are offered.

**Audience:**

Yes

**Claims And Evidence:**

Yes

**Requested Changes:**

Neither the methodology or the performance guarantee is really related to your problem (2). E.g., the heuristic should at least involve  $\epsilon$, $n$ and noise levels etc. to make the problem/solution more interesting. I believe some thoughts along this line will make the methods less incremental.

I also want to see why the existing of costs (compared with conservative/safe bandits literature) makes the problem particularly harder and why algorithms there cannot be extended - https://arxiv.org/abs/2111.04835 and https://arxiv.org/abs/2202.13234 are also relevant.

**Strengths And Weaknesses:**

Novelty: The novelty is incremental. While this specific problem formulation is for the first time been formulated in the context of pure exploration, the proposed algorithms are essentially heuristic adaptations of existing ones, lacking significant technical innovation.  See next section for more.

Quality: The methods are conceptually reasonable. Nonetheless, the numerical and theoretical justifications provided by the authors are not particularly robust, as detailed in subsequent sections. The theory is lack of much substance - under your bounded ratio assumption, it is more like copying the existing theoretical guarantees of the two base algorithms.

Clarity: The paper is well-structured and straightforward to understand.

Significance: The paper considers a practical application.

---

> ### Author Response · Authors · 2024-06-21
> **Response**
>
> - Thank you for your thoughtful feedback. We appreciate your encouraging remarks about our methods being conceptually reasonable, our work considering a practical application and the clarity of our presentation.
> - Novelty of the algorithms:
>   - We agree that the novelty of our algorithms is limited. Our contribution is in introducing the interesting problem setting of reducing cost to learn a good policy, and providing algorithms which show significant empirical improvements in reducing cost over prior methods. Our understanding is that our contribution is well aligned with the goals of TMLR evaluation criteria (https://jmlr.org/tmlr/editorial-policies.html) focused on “Are the claims made in the submission supported by accurate, convincing and clear evidence? Would at least some individuals in TMLR's audience be interested in knowing the findings of this paper? Papers should be accepted if they meet the criteria, even if the contribution or significance of the work is modest.”
>   - To better support our claims, we have included an additional experiment using a real-word dataset focused on mailing different types of letters to people to increase voter turnout (see details in the “experiment_on_voting_dataset.pdf” file in the Supplementary Material).
> - "Neither the methodology or the performance guarantee is really related to your problem: 2). E.g., the heuristic should at least involve $\epsilon$, $n$ and noise levels etc":
>   - For our first method, similar to Zanette et al. (2021), our heuristic does not include the parameter $\epsilon$ or noise levels, although it does affect sample complexity. We agree that the algorithm would be more robust if it also considered $\epsilon$. It would be straightforward to extend our algorithm to incorporate $\epsilon$. To do this, we could set $M$ in Algorithm 1 as the maximum number of samples needed, taking $\epsilon$ into account. Then, in Algorithm 2, as we sample in real-time, we would check at each time step whether the current covariance matrix is smaller than a certain amount. We would stop sampling once we hit the $\epsilon$ threshold. This approach would provide a clear stopping point while keeping the sampling policy non-adaptive. For simplicity, we assume $\eta$ is mean-zero, $1$-sub-Gaussian noise, so the noise level doesn’t appear in our algorithm. Since we need a static policy, the heuristic would not depend on $n$ (each round). We will update the paper accordingly.
>   - Both heuristics aim to have cost be directly considered as part of the algorithms.
>   - We agree that stronger theoretical results would be desirable. As we highlighted to reviewer i91Y, lower bounds suggest in the worst case, one cannot do much better than algorithms that aim to minimize the number of samples needed to learn a near optimal policy, with the resulting scaled amount of cost.
> - "why …costs (compared with conservative/safe bandits literature) makes the problem particularly harder":
>   - Thank you for the excellent suggestion. We will revise the related work and introduction to directly address this. Briefly, the conservative / safe bandits literature has primarily focused on minimizing cumulative regret, in MABs and CMABs, subject to some constraints on the policy class. For example, Amani et al 2019 instead consider cumulative regret with respect to the best safe actions, whereas we focus on reducing cost while learning a policy that minimizes simple regret. In Pacchiano et al. 2021 they do consider cost directly, but focus on minimizing cumulative regret while ensuring the deployed policy on each round satisfies a cost constraint, compared to our interest in minimizing cumulative cost to learn a good policy that minimizes simple regret. Similar to us, Carlsson et al 2024 focus on learning a near-optimal policy, but they consider multi-armed bandits with no context, and assume there are constraints on the arms. Work by Jamieson and others has considered how in some settings contextual MAB may be reduced to linear MABs, and an interesting direction for future work would be to consider if such results might allow a closer connection with our work (which considers contexts) and Carlsson et al. Note however that they focus on linear constraints on the policies themselves (through arm constraints) whereas we are interested in reducing cumulative cost.
>   - The two papers you cited by Kveton and colleagues (Zhu and Kveton; Wan, Kveton, Song) focus on policy evaluation of a single known policy $\pi$, subject to restrictions on the exploration policy $\pi_e$ being used to gather data to evaluate $V^{\pi}$, such as requirements for its value to have a sufficient value relative to the known performance of another policy. This is quite different than our setting in which we are learning a near-optimal policy and aim to reduce the total cost across episodes.
>   - Please just let us know if there are additional papers you would like us to compare to.

---

### Review · Reviewer_i91Y · 2024-06-10

**Summary Of Contributions:**

This paper studies pure exploration in resource-aware contextual bandits, where state-actions are not only associated with rewards but also different costs. The goal is to minimize the cumulative cost needed during exploration to learn an $\epsilon$-optimal policy.

The paper proposes two algorithms: one is adapted from [Zanette et al., 2021] for the static setting, where the exploration policy cannot be reactive to online data; and the other is adapted from [Russo and Van Roy, 2018] for the adaptive setting, where the policy can be updated. The algorithms maximize the uncertainty reduction/information gain *per unit cost*.

The paper evaluates the algorithms on synthetic and semi-synthetic data, shows that they use fewer resources than baseline algorithms that do not consider costs. The paper also shows that these adapted algorithms have sample complexity competitive with their resource-unaware counterparts.

**Audience:**

Yes

**Broader Impact Concerns:**

No particular concerns.

**Claims And Evidence:**

Yes

**Requested Changes:**

As discussed above, the paper can be much stronger if either empirical evidence on more real-world datasets or theoretical guarantees on the cumulative costs are given.

The presentation/clarity can also be improved:
- In the introduction, the motivating example can be clearer if you could explicitly specify what the contexts/actions/costs are.
- In Eq (4), do you mean $\Sigma_N^{-1}$?
- Do you assume, throughout the paper, that $\mathcal{A}$ is finite? I understand that you mentioned CMAB in Section 1, but can you clarify this formally in Section 3?
- In Section 5.1, what does $x_a$ represent for $a \in \{11, \ldots, 20\}$ when $\Sigma_a = \text{diag}(x_1,\ldots,x_{10})$?
- In Section 6, what is the exact definition of $\gamma$? The cost function $c$ seems to have been defined over state-action pairs.
- In Theorem 6.1, $\underline{m}$ is a variable in the algorithm and not defined here.
- Can you clarify when you consider the stateless setting in Section 6?

Some other papers that may be relevant:
- Amani, S., Alizadeh, M., & Thrampoulidis, C. (2019). Linear stochastic bandits under safety constraints. Advances in Neural Information Processing Systems, 32.
- Pacchiano, A., Ghavamzadeh, M., Bartlett, P., & Jiang, H. (2021, March). Stochastic bandits with linear constraints. In International conference on artificial intelligence and statistics (pp. 2827-2835). PMLR.
- Carlsson, E., Basu, D., Johansson, F., & Dubhashi, D. (2024, April). Pure exploration in bandits with linear constraints. In International Conference on Artificial Intelligence and Statistics (pp. 334-342). PMLR.

**Strengths And Weaknesses:**

Strengths:
- The studied problem seems relatively novel and interesting: regret minimization in contextual bandits with a resource budget/constraint has been studied before; this paper tackles a slightly different problem in which the goal is to identify an $\epsilon$-optimal policy using as little cost as possible.
- The paper presents two algorithms and provides empirical evidence from (semi)-synthetic data, demonstrating that the proposed algorithms require fewer resources. The paper also presents some theoretical evidence on the sample complexity of the algorithms.

Weaknesses:
- While the objective of the studied problem is interesting--*minimization* of the cumulative cost incurred during exploration, the paper does not really show that the cumulative cost is in fact minimized by the algorithms. Rather, the paper only shows two strategies that can *reduce* the cost during exploration. This is done through very minor modifications to existing algorithms in the literature [Zanette et al., 2021; Russo and Van Roy, 2018].
- In terms of the theoretical results, there are no guarantees on the total amount of resources incurred by these exploration algorithms; and there are also no lower bounds on how much cost is required. In fact, the provided guarantees can be somewhat vacuous, as they only show that the resource-aware algorithms do not require significantly more samples---which can also mean resources---than the resource-unaware ones.
- The experiments are conducted on a limited set of synthetic and semi-synthetic datasets. It would be interesting to see a more thorough empirical validation on more real-world datasets.

---

> ### Author Response · Authors · 2024-06-21
> **Response**
>
> - We thank the reviewer for their helpful feedback and encouraging remarks about the setting we consider.
> - “does not really show that the cumulative cost is in fact minimized by the algorithms”:
>   - We completely agree that we do not provide algorithms that are guaranteed to find the minimal cumulative cost. We view our primary contribution as providing a new important problem of how to reduce costs required to learn a good final decision policy, algorithms that directly aim to reduce costs, and experiments across synthetic, semi-synthetic, and real-world scenarios—including a new experiment using a voting dataset from Gerber et al., (2008). We will revise the introduction to be more precise in the motivating setting and our contributions in this paper.
> - no guarantees on the total amount of resources incurred by these exploration algorithms and no lower bounds on how much cost is required:
>   - Thanks for raising this important point. In the worst case, resource-efficient pure exploration for contextual bandits may be of similar hardness as sample-efficient pure exploration: as shown in \citet{chohlas2021learning}, if there is no information sharing across contexts and actions, acquiring information about the outcomes of a particular context-action pair will require sampling it directly. We will revise the paper to make this point clearer.
>   - We can provide a basic upper bound on the amount of resources incurred through a bound on the number of samples and known bounds on the cost, but in general this will depend on the contexts encountered. We agree that stronger bounds on the required budget to achieve a near-optimal policy would be a very interesting next step.
> - empirical evidence on more real-world datasets:
>   - We agree and have added a new experiment using a real-word dataset focused on mailing different types of letters to people to increase voter turnout in the 2006 Michigan primary election, collected by Gerber et al., (2008). We find that our cost-aware algorithm S-P_cost can learn a near-optimal decision policy using substantially fewer resources than traditional cost-unaware algorithms on the voting dataset as well (see details in the “experiment_on_voting_dataset.pdf” file in the Supplementary Material).
> - In the introduction, the motivating example can be clearer if you could explicitly specify what the contexts/actions/costs are.
>   - Thank you for the good suggestion, we will do so.
> - In Eq (4), do you mean $\Sigma_N^{-1}$?
>   - Yes. Thank you for catching this typo.
> - Do you assume, throughout the paper, that $\mathcal{A}$ is finite? I understand that you mentioned CMAB in Section 1, but can you clarify this formally in Section 3?
>   - Yes, we do assume the action space is finite and will add this.
> - In Section 5.1, what does $x_a$ represent for $a \in {11, \ldots, 20}$ when $\Sigma_a = \text{diag}(x_1,\ldots,x_{10})$?
>   - We meant $x_{a-10} = 1$. We will fix this.
> - In Section 6, what is the exact definition of $\gamma$? The cost function $c$ seems to have been defined over state-action pairs.
>   - Yes, $\gamma:= \frac{\max_{s,a} c(s,a)}{\min_{s,a} c(s,a)}$ should be defined over state and action. We will fix this.
> - In Theorem 6.1, $\underline{m}$ is a variable in the algorithm and not defined here.
>   - Thanks, we will fix this.
> - Can you clarify when you consider the stateless setting in Section 6?
>   - We have decided to remove the discussion of the stateless setting from the paper, as our contextual algorithms can be easily adapted to this setting. To avoid any confusion, we will delete this sentence from Section 6.
> - Some other papers that may be relevant:
>   - Thank you for bringing these to our attention. We will add a discussion to compare to each of these in the paper. Briefly, Amani et al 2019 instead consider cumulative regret with respect to the best safe actions, whereas we focus on reducing cost while learning a policy that minimizes simple regret. In Pacchiano et al. 2021 they do consider cost, but focus on minimizing cumulative regret while ensuring the deployed policy on each round satisfies a cost constraint, compared to our interest in minimizing cumulative cost to learn a good policy that minimizes simple regret. Similar to us, Carlsson et al 2024 focus on learning a near-optimal policy, but they consider multi-armed bandits with no context, and assume there are constraints on the arms. Work by Jamieson and others has considered how in some settings contextual MAB may be reduced to linear MABs, and an interesting direction for future work would be to consider if such results might allow a closer connection with our work (which considers contexts) and Carlsson et al. Note however that they focus on linear constraints on the policies themselves (through arm constraints) whereas we are interested in reducing cumulative cost.

---

### Decision · Action_Editor_5Fmg · 2024-07-13

**Recommendation:** Reject

**Comment:**

The paper received borderline acceptance and rejection recommendations. The topic is novel and warrants acceptance. However, the current paper feels more like a work in progress than a finished paper. The reviewers had two major comments:

* The proposed algorithms and theory are trivial extensions of prior works. While this may be true, it is not an acceptance criterion at TMLR.

* The paper feels incomplete for multiple reasons:

  * The main contribution of the paper is that it introduces an additional cost to pure exploration in contextual bandits. However, the incurred cost by the proposed algorithms is not analyzed. The bounds in Section 6 only bound the gap of the best empirical policy. They depend on the cost through a seemingly vacuous term, the maximum cost over the minimum cost.

  * The analysis is disconnected from the algorithms. As an example, while Theorem 6.1 depends on $\delta$ in the confidence interval width, Algorithm 1 does not have such a parameter. This is just a writing issue.

  * The error bounds in Section 6 may not be tight and this discounts their value. The tightness can be validated in several ways. Arguably the most common approach is to prove a matching lower bound. Another option is to show empirically that the gap of the best empirical policy scales as suggested by the bounds.

The above suggested changes go beyond a minor revision and therefore the paper cannot be accepted now.

I also suggest that the authors expand the discussion of prior works on bandits / MDPs with constraints. While this work does not use constraints, this is what popped up in the heads of all reviewers when they read the paper.

**Audience:**

This paper would be of a general interest to bandit and online learning communities.

**Claims And Evidence:**

Only partially. The theory is incomplete and disconnected from the algorithms. See my detailed comments.

**Resubmission Of Major Revision:**

The authors may consider submitting a major revision at a later time.